

# Psychological distress and pregnancy outcomes in early-stage gestational hypertension: a case-control study from China

Xilian Li[1],[*], Rongmin Wang[1],[*], Xian Xia[1], Dongdong Shi[1], Lili Gong[1] and Biao Gao[2]

[1] Obstetrics & Gynecology Hospital of Fudan University, Shanghai Key Lab of Reproduction and Development, Shanghai Key Lab of Female Reproductive Endocrine Related Diseases, Shanghai, China
[2] Teaching and Research Support Center, Naval Medical University, Shanghai, China
[*] These authors contributed equally to this work.

Corresponding author
Biao Gao, gbdata@163.com

## ABSTRACT

**Objective:** This study aimed to elucidate the pathophysiological role of psychological distress in early-stage gestational hypertension (GH) through comprehensive assessment of its regulatory effects on disease progression and association with adverse pregnancy outcomes, thereby providing evidence-based support for early screening and intervention strategies.

**Methods:** We conducted a prospective case-control study involving 446 patients with early-stage GH (diagnostic criteria: new-onset hypertension after 20 weeks of gestation, blood pressure ≥140/90 mmHg without proteinuria) and 200 normotensive pregnant women as controls. Psychological distress was assessed using the Self-Rated Anxiety Scale (SAS). A multidimensional statistical approach, including univariate analysis and multivariate logistic regression, was employed to systematically explore the risk factors that influence psychological distress. Pregnancy and perinatal outcomes were compared using Chi-square tests and t-tests.

**Results:** The study revealed a markedly elevated prevalence of psychological distress in the early-stage GH group (20.9%) compared to controls (7.0%, $P < 0.05$). Multivariate analysis identified educational level ($OR$ = 2.298, 95% CI [1.289–4.097]), history of adverse pregnancy ($OR$ = 2.604, 95% CI [1.342–5.050]), and GH itself ($OR$ = 1.859, 95% CI [1.213–2.850]) as independent risk factors for psychological distress. Follow-up data demonstrated that patients with psychological distress exhibited significantly higher rates of progression to preeclampsia (24.7% *vs.* 12.7%, $P < 0.05$), along with increased incidence of adverse pregnancy outcomes, including premature rupture of membranes, postpartum hemorrhage, neonatal infection, macrosomia, and low birth weight.

**Conclusions:** This study provides a systematic characterization of psychological distress patterns in early stage GH patients and their potential impact on disease progression. Findings highlight the critical importance of integrating routine psychological screening and early intervention strategies into prenatal care for patients with GH to optimize maternal and neonatal outcomes.

## SUMMARY

High blood pressure during pregnancy (gestational hypertension) is a common condition that may negatively impact the health of both mothers and their babies. Our study found that women with gestational hypertension were much more likely to experience anxiety. Anxiety can further complicate pregnancy, increasing the risks of preterm delivery, preeclampsia, postpartum bleeding, and babies with abnormal birth weight—including low birth weight and macrosomia. Our findings highlight the importance of mental health care during pregnancy. Doctors should routinely screen pregnant women for psychological distress and provide timely emotional support, especially for those diagnosed with high blood pressure, to help improve both maternal and neonatal outcomes.

## INTRODUCTION

Gestational hypertension (GH) is one of the most common medical conditions during pregnancy, defined as high blood pressure that occurs after 20 weeks of pregnancy without significant proteinuria. GH not only affects the health of pregnant women but is also associated with increased maternal and neonatal mortality and morbidity, as well as potential adverse health outcomes (*ACOG, 2019*). Previous studies have revealed a variety of risk factors, including, but not limited to, prepregnancy obesity, diabetes, advanced maternal age, and multiple pregnancies, all of which can contribute to the development of GH (*Bartsch et al., 2016*). These well-established biomedical predictors relate to the onset of GH; by contrast, the present study explores whether psychological distress arising after an early-stage GH diagnosis constitutes an additional, modifiable risk factor for poor maternal–fetal outcomes.

The importance of mental health during pregnancy is increasingly being recognized (*Shay et al., 2020*), particularly the potential impact of psychological distress on the health of pregnant women and their fetuses. In this study, psychological distress specifically refers to clinically significant anxiety, defined by a validated Self-Rating Anxiety Scale (SAS) score ≥50. This threshold reflects both cognitive-affective and somatic symptoms and has been validated in Chinese populations (*Xiujuan, 2021*; *Song et al., 2021*; *Jin et al., 2022*). Psychological distress is common during pregnancy and can negatively affect pregnancy outcomes by promoting sympathetic nervous system activity, increasing vasoconstriction, and increasing blood pressure (*Ding et al., 2014*).

Although studies have explored the relationship between GH and psychological distress, the results are inconsistent (*Katon et al., 2012*). The exact impact of psychological distress in GH patients on fetal development and pregnancy outcomes is still inconclusive (*Biaggi et al., 2016*; *Li et al., 2023*; *Pearlstein, 2015*). This may be related to variations in research methods, sample sizes, and assessment tools. We believe that by adopting more unified and rigorous assessment standards, we can more clearly reveal the association between these factors. Moreover, different sociocultural backgrounds may influence research outcomes.

Early stage psychological intervention in GH patients can help prevent disease progression and improve pregnancy outcomes. Clarifying the association between psychological distress and the clinical course of GH could therefore provide valuable insights for targeted management and intervention strategies (*Shay et al., 2020*; *Frayne et al., 2021*).

This observational study aims to answer the following key questions: (1) Do patients with GH exhibit significantly higher levels of psychological distress compared to healthy pregnant women? (2) What factors are associated with an increased risk of psychological distress in patients with GH? (3) How does psychological distress affect pregnancy outcomes in these patients?

Based on existing evidence, we hypothesize that the incidence of psychological distress is higher in patients with GH and that this distress is associated with adverse pregnancy outcomes. To validate this hypothesis, we designed a prospective case-control study utilizing both univariate and multivariate analyses, with detailed methodology provided in the Methods section.

## METHODS

### Clinical data

#### Sample size

We determined the sample size based on *a priori* estimation to obtain statistically significant results. Studies have reported a higher incidence of anxiety in GH patients (*Chapuis-de-Andrade et al., 2022*), approximately 26–32% (*Roberts, Davis & Homer, 2019*). Taking into account a significance level of 5%, 80% statistical power, and maximum sample loss, we planned to include 468 patients with GH and 215 control pregnant women as research subjects.

#### Recruitment and consent

Participants were recruited consecutively at the Obstetrics and Gynecology Hospital of Fudan University between June 6 2022 and June 20 2023. The institutional review board approved the study protocol. Before enrollment, all eligible women received a detailed explanation of the study's purpose, procedures, potential risks and benefits, as well as the voluntary nature of participation. They then provided written informed consent. To ensure valid self-reporting, women with diagnosed intellectual disability, dementia, or those who failed this checklist were excluded. Literacy was also confirmed to guarantee that participants could read and comprehend the questionnaires.

#### GH group

**Inclusion criteria:** (a) singleton pregnancy; (b) diagnosed with GH after 20 weeks of gestation, with blood pressure ≥140/90 mmHg; (c) no history of chronic diseases such as prepregnancy hypertension or diabetes; (d) no history of mental illness or substance abuse; (e) voluntary participation in the study and signed informed consent; (f) adequate cognitive capacity, confirmed as described in "Recruitment and Consent".

**Exclusion criteria:** (a) multiple pregnancies, (b) other pregnancy complications or medical/surgical conditions, (c) incomplete clinical data, and (d) participants that presented with or rapidly progressed to preeclampsia at initial diagnosis.

During the study period (6 June 2022 to 20 June 2023), a total of 7,970 first-visit pregnancies were recorded at our department. Of these, 468 women fulfilled the diagnostic criteria for early-stage GH and were consecutively screened. A total of 22 participants were excluded: 12 for failing to meet the eligibility criteria, four due to rapid progression to preeclampsia at presentation, and six for missing data, leaving 446 GH cases for the final analysis.

### Selection of controls

During the same period, 7,052 normotensive pregnancies served as the control pool. Using a computer-generated random sequence, we pre-selected 215 women (≈1:2 case-to-control ratio) and applied frequency matching on five a-priori confounders: maternal age (≤35 *vs.* >35 years), pre-pregnancy body mass index (BMI) (≤25 *vs.* >25 kg/m$^2$), parity (nulliparous/multiparous), educational level (high-school-or-below/college-or-above), and family history of hypertension (yes/no). After secondary screening, eight women failed the eligibility criteria and seven had incomplete records; hence, 200 controls were included in the final analysis.

## Research methods

### Data collection

A self-designed questionnaire was used to collect demographic characteristics of study subjects, such as age, level of education, occupation, obstetric history, family history, blood pressure, body mass index, and other clinical data. Based on literature and clinical relevance, the following variables were prespecified as potential confounders and recorded for every participant: maternal age, pre-pregnancy BMI, educational level, occupation, parity, adverse pregnancy history (≥1 miscarriage, stillbirth or preterm birth), family history of hypertension (diagnosis of hypertension in a first-degree relative), and comorbid thyroid disease. All these variables were recorded during data collection and included in the analysis.

We categorized certain continuous variables according to clinical or epidemiological significance. Age was dichotomized using 35 years as the cut-off point, resulting in two categories: ≤35 years and >35 years. This classification was chosen because advanced maternal age (typically defined as >35 years) is a recognized obstetric risk factor. Pre-pregnancy BMI was dichotomized using 25 kg/m$^2$ as the cut-off point, resulting in two categories: ≤25 and >25. This categorization was selected because a BMI > 25 is defined as overweight or obese, which is associated with pregnancy complications and adverse outcomes.

Missing data were handled using complete-case analysis. A total of 13 participants (six in the GH group and seven in the control group), representing 1.9% of the estimated total sample, were excluded from the multivariable modeling due to missing data for one or more variables.

### Flow chart

The flow chart for the recruitment and screening is as follows (Fig. 1):

### Assessment of psychological distress

Psychological distress assessment was conducted using the widely used SAS (*Zung, 1971*), which was selected based on their demonstrated applicability in evaluating the psychological distress of pregnant women in multiple studies and their ability to provide quantitative scores, facilitating statistical analysis (*Dunstan, Scott & Todd, 2017*). The scales comprise 20 items, covering both cognitive-affective (items 1–5, 11–15) and somatic (items 6–10, 16–20) symptoms, with a four-level scoring system (score range: 20–80 points). Based on validated Chinese norms, we adopted a cutoff value of 50 points for SAS (*Xiujuan, 2021*; *Song et al., 2021*; *Jin et al., 2022*). All participants independently completed the questionnaire in approximately 5 to 10 min, between 24 and 32 weeks of gestation, in a private consultation room at the hospital, ensuring a standardized environment for data collection. The standardized assessment procedure was strictly maintained by trained research assistants to minimize response bias. Immediately after completion, the questionnaires were scored, and the scores were entered into a database for further analysis. For participants with visual impairments, research assistants read each item aloud *verbatim* without additional prompting, ensuring that all participants were able to complete the questionnaire according to standard procedures.

Primarily serving as an anxiety scale, the Zung SAS has demonstrated correlations with depressive symptoms (*Shao et al., 2020*). Given this overlap, the tool may also indirectly capture certain aspects of depression. It has shown consistent reliability and validity in assessing psychological distress across various stages of pregnancy in different cultural contexts (*Dong et al., 2021*; *Kang et al., 2016*; *Jalal, Alsebeiy & Alshealah, 2024*). This makes it a versatile and widely used tool in obstetric research. We specifically chose to use the SAS to assess anxiety because anxiety is the most prevalent psychological distress during pregnancy, particularly in women diagnosed with GH, and it is crucial to identify targeted interventions for this condition (*Yan & Zhou, 2025*).

### Evaluation of pregnancy outcomes

Two groups of pregnant women were followed and their pregnancy results were recorded, including pregnancy complications (gestational diabetes, preeclampsia, placental abruption, *etc.*), mode of delivery (vaginal delivery or cesarean section), preterm delivery (between 28 and 37 weeks of gestation), neonatal conditions (low birth weight, neonatal asphyxia, *etc.*) and postpartum hemorrhage.

### Statistical methods

Statistical analyses were performed using SPSS version 26.0 (IBM Corp., Armonk, NY, USA). Based on study design and data characteristics, measurement data were presented as mean ± standard deviation ($\bar{x} \pm s$), with intergroup differences examined using *t*-tests after confirming normality of distribution. For categorical variables, we employed Chi-square tests to analyze frequency distributions and percentages between groups. Multivariate

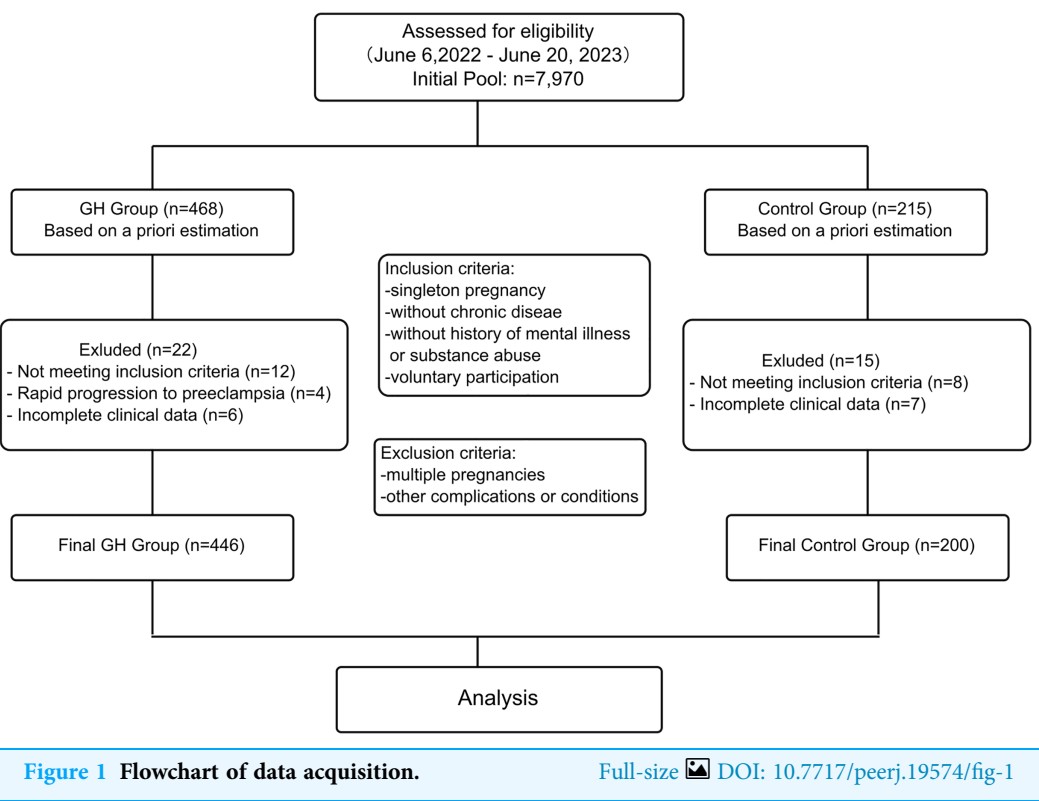

**Figure 1 Flowchart of data acquisition.**

logistic regression analysis was employed to elucidate independent risk factors associated with psychological distress, with statistical significance established at $P < 0.05$.

To address potential confounding factors, we developed an analytical strategy based on comprehensive literature review and clinical observations. Key factors influencing anxiety and pregnancy outcomes were identified and evaluated to determine essential adjustment variables. Variables demonstrating potential significance ($P < 0.10$) in univariate analysis, including thyroid disease, were incorporated into the initial multivariate model, which was refined through backward elimination (exit criteria: $P > 0.05$) to optimize precision and interpretability. Strategic matching design and stratification methods further enhanced our approach ; however, the stratified analysis within the control group was inconclusive owing to limited sample size and is therefore not presented.

After data collection, a retrospective power check was planned to verify whether the achieved sample size met the initial assumptions, using the formula $n = (Z\alpha/2 + Z\beta)^2 \times [P1(1 - P1) + P2(1 - P2)]/(P1 - P2)^2$ (*Sakpal, 2010*; *Fleiss, Levin & Paik, 2003*).

### Follow-up and outcome ascertainment

Participants were followed from enrolment at a mean gestational age of 24 ± 3 weeks until delivery (38 ± 2 weeks). Because all participants delivered at the study hospital, primary maternal–fetal outcomes were documented immediately after delivery; therefore, no post-discharge follow-up was required. Automated linkage between outpatient registration

numbers and inpatient delivery records ensured complete outcome capture, and no loss to follow-up occurred.

### Ethics approval

The Research Ethics Committee of the Fudan University Obstetric and Gynecology Hospital reviewed and approved the research protocol. All aspects of this study were conducted according to the guidelines of Obstetrics and Gynecology (Ref No: EROG2022-102). Details of the informed consent process are described in "Recruitment and Consent".

## RESULTS

### Baseline characteristics

The baseline characteristic analysis did not reveal significant differences ($P > 0.05$) between the control group and the GH group in terms of general information, including age, BMI before pregnancy, level of education, occupation, parity, and family history of hypertension (Table 1). The two groups were comparable in baseline characteristics. This sampling method allowed us to comprehensively assess the incidence of psychological distress in GH patients and evaluate the impact of these psychological states on pregnancy outcomes.

### Comparison of psychological distress between two groups

In the group of pregnant women with GH, 20.9% of them experienced psychological distress, while in the control group, 7.0% of pregnant women experienced psychological distress. The proportion of pregnant women with psychological distress in the GH group was significantly higher than in the control group, and the difference was statistically significant ($\chi^2 = 19.17$, $P < 0.05$) (Table 1).

According the observed incidence of psychological distress in our study (20.9% in the GH group and 7.0% in the control group), retrospective power analysis confirmed that the final sample (446 cases + 200 controls) exceeded the minimum required size of 113 per group, ensuring 80% power at $\alpha = 0.05$.

### Univariate analysis

Univariate analysis showed that the differences in incidence of psychological distress was statistically significant ($P < 0.05$) in terms of different levels of education, whether there was a history of adverse pregnancy outcomes, the presence of GH and the presence of concomitant thyroid disorder (Table 2).

### Multivariate logistic regression analysis

Based on indicators with statistical significance ($P < 0.10$) in the univariate analysis, we performed a multivariate logistic regression analysis with the appearance of psychological distress as the dependent variable. The criteria for selecting adjustment factors were: (1) the previous literature indicates their association with anxiety and pregnancy outcomes; (2) the preliminary analysis showed that they were potential confounders of exposure and outcome; (3) their association with psychological distress meeting the preset significance level in the univariate analysis. We used a backward selection method

**Table 1 Baseline characteristics and psychological distress status of participants.**

| | | Control group (n = 200) | GH group (n = 446) | t/χ² | P |
|---|---|---|---|---|---|
| Age (years) | | 29.52 ± 2.69 | 30.57 ± 8.09 | −1.79 | 0.075 |
| Pre-pregnancy BMI (kg/m²) | | 22.63 ± 3.02 | 21.84 ± 6.81 | −1.57 | 0.118 |
| Education level | High school/Technical school and below | 38 (19.00%) | 80 (17.94%) | 0.045 | 0.831 |
| | College and above | 162 (81.00%) | 366 (82.06%) | | |
| Occupation | Employed | 142 (71.00%) | 334 (74.89%) | 0.885 | 0.347 |
| | Unemployed | 58 (29.00%) | 112 (25.11%) | | |
| Parity | Primiparous | 162 (81.00%) | 343 (76.91%) | 1.127 | 0.288 |
| | Multiparous | 38 (19.00%) | 103 (23.09%) | | |
| Family history of hypertension | Yes | 2 (1.00%) | 6 (1.35%) | OR 0.74* | 1.000 |
| | No | 198 (99.00%) | 440 (98.65%) | | |
| Psychological distress | Yes | 14 (7.00%) | 93 (20.85%) | 18.182 | 0.000 |
| | No | 186 (93.00%) | 353 (79.15%) | | |

**Note:**
*Fisher's exact test was used due to expected frequencies <5 in certain cells.

**Table 2 Univariate analysis of factors influencing psychological distress.**

| | | Psychological distress (n = 107) | Non-psychological distress (n = 539) | χ² | P |
|---|---|---|---|---|---|
| Age (years) | ≤35 | 81 (75.70%) | 425 (78.85%) | 0.353 | 0.553 |
| | >35 | 26 (24.30%) | 114 (21.15%) | | |
| Education level | High school/Technical school and below | 30 (28.04%) | 88 (16.32%) | 7.44 | 0.006 |
| | College and above | 77 (71.96%) | 451 (83.67%) | | |
| Pre-pregnancy BMI (kg/m²) | ≤25 | 85 (79.44%) | 450 (83.49%) | 0.764 | 0.382 |
| | >25 | 22 (20.56%) | 89 (16.51%) | | |
| Occupation | Employed | 81 (%) | 395 (73.28%) | 0.159 | 0.690 |
| | Unemployed | 26 (%) | 144 (26.72%) | | |
| Parity | Primiparous | 88 (82.24%) | 417 (77.37%) | 0.975 | 0.323 |
| | Multiparous | 19 (17.76%) | 122 (22.63%) | | |
| Adverse pregnancy history | Yes | 33 (30.84%) | 103 (19.11%) | 6.704 | 0.010 |
| | No | 74 (69.16%) | 436 (80.89%) | | |
| GH | Yes | 93 (86.92%) | 353 (65.49%) | 18.182 | 0.000 |
| | No | 1,413.08 (%) | 186 (34.51%) | | |
| Thyroid disease | Yes | 15 (14.02%) | 38 (7.05%) | 4.869 | 0.027 |
| | No | 92 (85.98%) | 501 (92.95%) | | |
| Family history of hypertension | Yes | 1 (0.93%) | 7 (1.30%) | 0.000 | 1.000 |
| | No | 106 (99.07%) | 532 (98.70%) | | |

($P > 0.05$) to obtain the final model, balancing the adequacy of the control of the confounder factor and model parsimony.

The results of the multivariate logistic regression analysis (Table 3) showed that education level, adverse pregnancy history, and GH were independent risk factors for

**Table 3 Multivariate analysis of factors influencing psychological distress.**

| | | β | SE | Wald χ² | P | OR | 95% CI |
|---|---|---|---|---|---|---|---|
| Education level | High school/Technical school and below | 0.832 | 0.295 | 7.954 | 0.012 | 2.298 | [1.289–4.097] |
| | College and above | 1.036 | 0.315 | 10.816 | 0.007 | 2.818 | [1.520–5.225] |
| Adverse pregnancy history | | 0.957 | 0.338 | 8.016 | 0.000 | 2.604 | [1.342–5.050] |
| Coexisting GH | | 0.620 | 0.218 | 8.088 | 0.033 | 1.859 | [1.213–2.850] |

**Note:**
Thyroid disease, initially significant at univariate analysis ($P < 0.05$), was entered into the logistic regression model but was excluded during the backward elimination procedure ($P > 0.05$ at step 2), indicating no independent predictive value when adjusted for other covariates.

psychological distress. Precisely: (1) Low education level: $OR = 2.298$, 95% CI [1.289–4.097]. (2) Adverse pregnancy history: $OR = 2.604$, 95% CI [1.342–5.050]. (3) GH: $OR = 1.859$, 95% CI [1.213–2.850]. After adjusting for other factors, pregnant women with low education levels, an adverse pregnancy history and GH had 2.298-, 2.604-, and 1.859-times higher risk of psychological distress, respectively, compared to their counterparts without these risk factors.

## Comparison of pregnancy outcomes

All pregnant patients with hypertension during pregnancy were divided into a psychological distress group and a non-psychological distress group based on the presence of psychological distress. Baseline data from the two groups were compared, and only the systolic and diastolic blood pressures of the psychological distress group were significantly higher than those of the non-psychological distress group (Table 4). The results of delivery of pregnant women and the perinatal outcomes of infants were compared between the psychological distress group and the non-psychological distress group. The results showed that in terms of the results of pregnant women, the incidence of preterm delivery, premature rupture of membranes, preeclampsia, and postpartum hemorrhage in the psychological distress group was significantly higher than in the non-psychological distress group ($P < 0.05$). In terms of perinatal outcomes, the incidence of neonatal infection, macrosomia, and low birth weight babies in the psychological distress group were also significantly higher than in the non-psychological distress group ($P < 0.05$) (Table 5).

## DISCUSSION

### Psychological distress in GH

Although GH generally has a better prognosis than preeclampsia, early identification and management of GH remain important as some cases may progress to more severe forms of hypertensive disorders of pregnancy (ACOG, 2019; Mol et al., 2016). Understanding the psychological distress of GH patients and their potential influence on disease progression could provide information for early intervention strategies (Reddy & Jim, 2019). According to a report by the World Health Organization, about 10% of pregnant women are at risk for GH, and up to 4% of maternal deaths are related to GH (WHO, 2011). This study found that the incidence of psychological distress in GH patients (20.9%) was significantly higher than that of ordinary pregnant women, similar to the results of Jinling, Jialan & Peipei

**Table 4 Comparison between psychological distress and non-psychological distress group.**

| | | Psychological distress (*n* = 93) | Non-psychological distress (*n* = 353) | t/χ² | *P* |
|---|---|---|---|---|---|
| Age (years) | | 29.53 ± 3.87 | 29.82 ± 3.52 | 0.660 | 0.510 |
| Pre-pregnancy BMI (kg/m²) | | 22.38 ± 3.23 | 21.86 ± 3.53 | 1.360 | 0.175 |
| Blood pressure | Systolic blood pressure | 152.37 ± 34.79 | 141.65 ± 28.63 | 2.740 | 0.006 |
| | Diastolic blood pressure | 105.78 ± 16.57 | 85.76 ± 15.98 | 10.440 | 0.000 |
| Education level | High school/Technical school | 18 (19.35%) | 62 (17.56%) | 0.160 | 0.688 |
| | College and above | 75 (80.65%) | 291 (82.44%) | | |
| Occupation | Employed | 65 (69.89%) | 269 (76.20%) | 1.559 | 0.211 |
| | Unemployed | 28 (30.11%) | 84 (23.80%) | | |
| Parity | Primiparous | 69 (74.19%) | 274 (77.62%) | 0.313 | 0.580 |
| | Multiparous | 24 (25.81%) | 79 (22.38%) | | |
| Family history of hypertension | Yes | 2 (2.15%) | 4 (1.13%) | 0.063 | 0.801 |
| | No | 91 (97.85%) | 349 (98.87%) | | |

**Table 5 Pregnancy outcomes differences in psychological distress groups.**

| | | Psychological distress (*n* = 93) | Non-psychological distress (*n* = 353) | χ² | *P* | OR (95% CI) |
|---|---|---|---|---|---|---|
| Delivery complications | Preterm delivery | 19 (20.43%) | 43 (12.18%) | 4.538 | 0.033 | 1.85 [1.03–3.32] |
| | Polyhydramnios | 13 (13.98%) | 38 (10.76%) | 0.775 | 0.379 | 1.35 [0.68–2.69] |
| | Premature rupture of membranes | 22 (23.66%) | 51 (14.45%) | 4.497 | 0.034 | 1.84 [1.05–3.22] |
| | Preeclampsia | 23 (24.73%) | 4 (12.75%) | 8.334 | 0.004 | 2.25 [1.27–3.99] |
| | Postpartum hemorrhage | 11 (11.83%) | 16 (4.53%) | 6.888 | 0.009 | 2.82 [1.26–6.31] |
| | Cesarean section | 42 (45.16%) | 173 (49.01%) | 0.430 | 0.512 | 0.86 [0.54–1.37] |
| Perinatal outcomes | Fetal distress | 9 (9.68%) | 19 (5.38%) | 2.307 | 0.129 | 1.89 [0.82–4.35] |
| | Neonatal infection | 10 (10.75%) | 11 (3.12%) | 9.231 | 0.002 | 3.75 [1.55–9.06] |
| | Macrosomia | 6 (6.45%) | 3 (0.85%) | 8.478 | 0.004 | 7.98 [1.95–32.70] |
| | Low birth weight | 9 (9.68%) | 14 (3.97%) | 4.883 | 0.027 | 2.59 [1.09–6.17] |

Note:
Premature membrane rupture refers to premature rupture of the fetal membranes rupture prematurely before delivery. Postpartum hemorrhage refers to bleeding of 500 ml or more for vaginal delivery and 1,000 ml or more for cesarean delivery within 24 h after birth. Fetal distress is an emergency state characterized mainly by hypoxemia and acidosis caused by fetal hypoxia in the uterus. Neonatal infection refers to diseases such as respiratory system infection, digestive system infection, central system infection, and blood infection in newborns. Macrosomia refers to a birth weight greater than 4,500 g, while low birth weight refers to a newborn weighing less than 2,500 g.

(2023) (22.5%). Further analysis showed that low level of education (*OR* = 2.298), adverse pregnancy history (*OR* = 2.604), and coexisting GH (*OR* = 1.859) were significant independent risk factors for psychological distress among pregnant women. Pregnant women with low levels of education may lack psychological resources to cope with pregnancy stress (*Changjia et al., 2020*); those with an adverse pregnancy history may have excessive concerns about pregnancy safety and fetal health, increasing their psychological burden; and regular medical monitoring required by GH itself can also increase the psychological pressure of patients (*Raina et al., 2021*). These results suggest that improving the educational level of pregnant women, improving psychological health screening for

high-risk pregnancies (such as those with an adverse pregnancy history), and optimizing the management of early GH may help reduce the incidence of psychological distress.

## Psychological distress on outcomes of GH

Psychological distress not only increases the burden on patients with early GH but may also influence disease progression and pregnancy outcomes (*O'Donnell et al., 2023*; *Frayne et al., 2021*). This study found that the systolic and diastolic blood pressure of GH patients with psychological distress showed significantly higher systolic and diastolic blood pressure than those without psychological distress. Furthermore, they demonstrated higher rates of progression to preeclampsia (24.7% *vs*. 12.7%), as well as increased incidence of premature delivery, postpartum hemorrhage, and low birth weight infants. Studies by *Winkel et al. (2015)* and *Shay et al. (2020)* also support the relationship between psychological distress and hypertensive disorders of pregnancy.

The pathophysiological mechanisms underlying this association are complex and multidimensional. Psychological distress can modulate gestational hypertension through interconnected neuroendocrine and immunological pathways: by dysregulating the hypothalamic-pituitary-adrenal (HPA) axis, elevating cortisol levels, and triggering chronic inflammatory processes. These physiological alterations can induce vasoconstriction, promote inflammatory factor release, disrupt placental development, and compromise fetal growth. Consequently, negative psychological states may not only escalate complication risks but also fundamentally alter the maternal-fetal physiological environment, potentially explaining the observed increased severity of hypertensive disorders (*Chu et al., 2024*).

## Clinical strategies and implications

Our findings revealed a complex, bidirectional relationship between psychological distress and GH. Psychological distress may exacerbate GH severity through neuroendocrine and inflammatory pathways, while GH itself significantly elevates anxiety level. The increased rates of preeclampsia progression (24.7% *vs*. 12.7%) in patients with psychological distress suggest potential biological mechanisms linking mental state to disease severity. One possible mechanism involves activation of the hypothalamic-pituitary-adrenal axis, leading to elevated cortisol levels and subsequent vascular dysfunction. Additionally, psychological distress may impact treatment compliance and lifestyle modifications, indirectly affecting disease management (*Chinese Preventive Medicine Association Psychosomatic Health Group Women's Mental Health Technology Group of China Maternal and Child Health Association, 2019*; *Frayne et al., 2021*).

The observed association between educational level and psychological distress (*OR* = 2.298, 95% CI [1.289–4.097]) points to the importance of tailoring patient education and communication strategies. Furthermore, the identification of adverse pregnancy history as a significant risk factor (*OR* = 2.604, 95% CI [1.342–5.050]) highlights the need for specialized care protocols for this vulnerable subgroup.

These mechanistic insights and risk factor analyses provide a foundation for developing targeted intervention strategies, potentially improving the precision of prenatal care in GH patients.

### Limitations and strengths

**Limitations:** This investigation was conducted at a single tertiary hospital, which may restrict the generalisability of the findings to other settings or ethnic groups. Owing to case-control design, the study cannot establish causal relationships between psychological distress and pregnancy outcomes. Although we controlled for several key confounders, unmeasured variables—such as social support networks and detailed socioeconomic indicators—could still bias the associations. Complete-case analysis, while practical, may have introduced selection bias. In addition, we were unable to monitor the dynamic interplay between changes in psychological status and blood-pressure trajectories across gestation, nor to delineate the precise timing of progression from early-stage GH to preeclampsia. A parallel subgroup analysis in normotensive controls was attempted but proved under-powered.

**Strengths:** Despite these constraints, the study possesses several noteworthy strengths. First, sample size (446 patients with GH and 200 healthy controls) provided adequate statistical power, and psychological distress was measured with the locally validated SAS, guaranteeing cultural applicability and data comparability (*Dunstan, Scott & Todd, 2017*). Second, frequency matching on five a-priori confounders and a dual analytical strategy (univariate screening followed by multivariable logistic regression) enhance internal validity (*Dennis, Janssen & Singer, 2004*; *Sperandei, 2014*). Third, maternal-fetal outcomes were recorded prospectively and immediately after delivery, providing robust clinical endpoints that underscore the relevance of incorporating routine psychological evaluation into GH management.

## CONCLUSIONS

This comprehensive case-control study provides insights into the relationship between psychological distress and early-stage GH. The significantly higher prevalence of psychological distress in GH patients (20.9% *vs.* 7.0% in controls), coupled with its association with disease progression and adverse pregnancy outcomes, emphasizes the importance of psychological assessment in prenatal care. The identification of education level, adverse pregnancy history, and GH as independent risk factors offers valuable guidance for targeted screening. Our findings suggest that incorporating routine psychological evaluation and support into the standard management protocol for early-stage GH could potentially improve maternal-fetal outcomes. This study represents the comprehensive systematic investigation elucidating the psychological distress profiles among early-stage pregnancy-induced hypertension patients, while critically examining their potential implications for disease progression. Our findings not only provide substantive evidence for early clinical screening and targeted interventions but also

underscore the imperative for future research to rigorously evaluate the efficacy of psychological intervention strategies in modulating pregnancy outcomes.

Future multicenter studies investigating the effectiveness of early psychological interventions in preventing disease progression are warranted.

## ACKNOWLEDGEMENTS

We would like to express our sincere gratitude to colleagues from Naval Medical University and the Obstetrics and Gynecology Hospital of Fudan University for their valuable contributions to this research. We especially acknowledge the Obstetrics and Gynecology Hospital of Fudan University for providing the necessary research facilities and administrative support.

### Funding

The work was supported by the Military Key Discipline Construction Projects of China (HL21JD1206) and the China Medical Board (CMB21-428). The funders had no role in study design, data collection and analysis, decision to publish, or preparation of the manuscript.

### Grant Disclosures

The following grant information was disclosed by the authors:
Military Key Discipline Construction Projects of China: HL21JD1206.
China Medical Board: CMB21-428.

### Competing Interests

The authors declare that they have no competing interests.

### Author Contributions

- Xilian Li conceived and designed the experiments, performed the experiments, prepared figures and/or tables, authored or reviewed drafts of the article, and approved the final draft.
- Rongmin Wang performed the experiments, authored or reviewed drafts of the article, and approved the final draft.
- Xian Xia analyzed the data, authored or reviewed drafts of the article, and approved the final draft.
- Dongdong Shi performed the experiments, authored or reviewed drafts of the article, and approved the final draft.
- Lili Gong analyzed the data, authored or reviewed drafts of the article, and approved the final draft.
- Biao Gao conceived and designed the experiments, prepared figures and/or tables, authored or reviewed drafts of the article, and approved the final draft.

## Human Ethics

The following information was supplied relating to ethical approvals (*i.e.*, approving body and any reference numbers):

The Research Ethics Committee of the Fudan University Obstetric and Gynecology Hospital granted Ethical approval to carry out the study within its facilities (Ethical Application Ref: EROG2022-102).

## Data Availability

The raw measurements are available in the Supplemental Files.

## Supplemental Information

Supplemental information for this article can be found online at http://dx.doi.org/10.7717/peerj.19574#supplemental-information.

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
