# Peer review of "Psychological distress and pregnancy outcomes in early-stage gestational hypertension: a case-control study from China"

_PeerJ, doi:10.7717/peerj.19574_

## Round 0.1 · original submission · Major Revisions

· Academic Editor

Major Revisions

Dear authors,

Thank you for your submission.

Improvements are needed to enhance the clarity of the methodological approach and data analysis. Please, refer to the reviewers' comments for further details.

Reviewer 1 ·

Basic reporting

The paper lacks clarity. The lack of clarity is partly due to grammatical problems. However, it also suffers lack of clarity in the method and material section, or in developing hypotheses. I have put comments in the PDF version of the submitted paper. An example is that they have not elaborated on the confounding variable, although they have said they reviewed literature and used appropriate statistical tests to control for the confounding variables.

Experimental design

The research questions are clear. However, the method and material section is not. I have put my comments in the pdf file.

Validity of the findings

Reviewing the supplemental material did actually increase my confusion. There were inconsistencies, at least to my understanding, regarding the numbers of the cases (I have commented on it in the PDF file). The hypothesis of the impact of psychological problems on negative pregnancy outcomes needs clarification, as the authors have mentioned that GH is a risk factor for psychological distress and GH has a negative impact on pregnancy outcomes. How did they control for the impact of GH on negative outcomes when it is a risk factor for psychological distress itself?

Annotated reviews are not available for download in order to protect the identity of reviewers who chose to remain anonymous.

Reviewer 2 ·

Basic reporting

The manuscript was well-written in English, with sufficient literary references and contextual support. It was also well-formatted, and the hypothesis and objectives were clearly stated.

Experimental design

1. How were the cases and controls matched? You mentioned key potential confounding factors—please specify which confounders were used for matching and describe the matching process (e.g., frequency matching or exact matching). Additionally, how were the cases selected? Were all patients with GH during the study period included, or was a random selection method used? If random selection was applied, please provide the total number of GH patients admitted during that period to establish the basis for selection. Furthermore, Figure 1 lacks information regarding the initial (larger) pool of patients from which the cases and controls were drawn.

2. Why was the SAS score used to assess psychological distress instead of other scoring methods? Does SAS offer specific advantages, or was its use based on prior studies? Please clarify.

3. Please specify the follow-up duration from inclusion to delivery. As reported, there was no loss to follow-up—can you confirm if this is accurate? If so, how was complete follow-up ensured?

Validity of the findings

1. Please merge Tables 1 and 2 into a single table, as separating one finding into Table 2 is unnecessary.

2. In Table 4, according to your statistical approach (including covariates with p<0.1 in the multivariable regression), please clarify why thyroid disease was not included in the regression analysis despite meeting this criterion. In the case that thyroid disease fell out while using backward stepwise regression, please clarify.

3. Please specify in the titles of Tables 5 and 6 that they present a subgroup analysis of patients with GH. Additionally, could you perform a similar analysis for the control group (patients without GH) and compare the findings with the GH group?

4. The content in lines 197 to 201 should be moved to the "Methods" section, as it does not belong in the "Results" section.

5. Section 4.1 should be more concise and integrated with the "Limitations" section to form a new section titled "Limitations and Strengths."

---

## Round 0.2 · accepted · Accept

· Academic Editor

Accept

Dear authors,

Both reviewers are happy with the revisions conducted. I am now accepting your manuscript for publication. congratulations!

Reviewer 1 ·

Basic reporting

Have no further comments after the changes have been made.

Experimental design

Have no further comments after the changes have been made.

Validity of the findings

Have no further comments after the changes have been made.

Additional comments

Have no further comments after the changes have been made.

Reviewer 2 ·

Basic reporting

This revised manuscript was very well-written and easy to understand. All of my comments were addressed appropriately. I have no further suggestion. This manuscript is ready to be published.

Experimental design

The study design was improved accordingly.

Validity of the findings

The findings were valid and support the conclusion.